# Genomic Analysis of an I1 Plasmid Hosting a *sul3*-Class 1 Integron and *bla*_SHV-12_ within an Unusual *Escherichia coli* ST297 from Urban Wildlife

**DOI:** 10.3390/microorganisms10071387

**Published:** 2022-07-10

**Authors:** Ethan R. Wyrsch, Monika Dolejska, Steven P. Djordjevic

**Affiliations:** 1Australian Institute for Microbiology & Infection, University of Technology Sydney, Ultimo, NSW 2007, Australia; ethan.wyrsch@uts.edu.au; 2CEITEC VETUNI, University of Veterinary Sciences Brno, 61242 Brno, Czech Republic; monika.dolejska@gmail.com; 3Department of Biology and Wildlife Diseases, Faculty of Veterinary Hygiene and Ecology, University of Veterinary Sciences Brno, 61242 Brno, Czech Republic; 4Department of Clinical Microbiology and Immunology, Institute of Laboratory Medicine, University Hospital Brno, 62500 Brno, Czech Republic

**Keywords:** AMR, antibiotic, IS*26*, Tn*21*, extended spectrum β-lactamase, *Escherichia coli*

## Abstract

Wild birds, particularly silver gulls (*Chroicocephalus novaehollandiae*) that nest near anthropogenic sites, often harbour bacteria resistant to multiple antibiotics, including those considered of clinical importance. Here, we describe the whole genome sequence of *Escherichia coli* isolate CE1867 from a silver gull chick sampled in 2012 that hosted an I1 pST25 plasmid with *bla*_SHV-12,_ a β-lactamase gene that encodes the ability to hydrolyze oxyimino β-lactams, and other antibiotic resistance genes. Isolate CE1867 is an ST297 isolate, a phylogroup B1 lineage, and clustered with a large ST297 O130:H11 clade, which carry Shiga toxin genes. The I1 plasmid belongs to plasmid sequence type 25 and is notable for its carriage of an atypical *sul3-*class 1 integron with *mefB*_∆260_, a structure most frequently reported in Australia from swine. This integron is a typical example of a Tn*21*-derived element that captured *sul3* in place of the standard *sul1* structure. Interestingly, the mercury resistance (*mer*) module of Tn*21* is missing and has been replaced with Tn*2*-*bla*_TEM-1_ and a *bla*_SHV-12_ encoding module flanked by direct copies of IS*26*. Comparisons to similar plasmids, however, demonstrate a closely related family of ARG-carrying plasmids that all host variants of the *sul3*-associated integron with conserved Tn*21* insertion points and a variable presence of both *mer* and *mefB* truncations, but predominantly *mefB*_∆260_.

## 1. Introduction

Antibiotics have been used to control infectious agents in human and veterinary medicine and in agriculture since the 1930s and underpin modern infection control strategies. Although resistance to antibiotics is a natural phenomenon, the scale and rate of resistance has risen over the past 100 years, presumably driven by human activity.

The environmental accumulation of expired and unmetabolized antibiotics, plus other pharmaceutical waste, is of particular concern when looking at the persistence and evolution of antimicrobial resistance [1,2]. Environmental antimicrobial residues have been detected in over 70 countries worldwide, underscoring severe limitations in the current pollution mitigation strategies. Sulphonamides, macrolides, quinolones and tetracyclines are notable pollutants in this regard [3]. While animal production is considered a leading contributor to environmental antibiotic and metal residues, waste from hospitals, assisted living facilities, municipal and industrial waste sites contribute significantly. Subinhibitory concentrations of antibiotics in sediments and aquatic environments are considered sufficient to influence microbial ecology and drive antibiotic resistance [4].

Sulphonamides have a long and varied history, serving as a key antibacterial agent during the development of antimicrobial chemotherapy [5]. However, the effectiveness of this drug family in *Enterobacteriaceae* was compromised by the widespread dissemination of the class 1 integron. There a four well described sulphonamide resistance genes found in *Escherichia coli*, including *sul1*, *sul2*, *sul3* and *sul4.* All four *sul* genes are localized on mobile genetic elements and have an association with the class 1 integron, a DNA recombination system that captures and expresses a myriad of antimicrobial resistance genes (ARGs). 

The sustained use of sulphonamides, co-trimoxazole (a combination treatment comprised of sulphamethoxazole and trimethoprim) and antiseptics has almost certainly influenced the genetic composition of the clinical class 1 integron, a key element that harbors ARGs and is a core component of larger genetic structures known as complex resistance regions (CRRs). The structure of the archetypal clinical class 1 integron includes a 5′ conserved segment (5′-CS) comprising the integron integrase (*intI1*) and a 3′-CS comprising the sulphonamide resistance gene *sul1* and a fused but functional biocide resistance gene *qacE∆1*. Between the 5′-CS and the 3′-CS resides a variable region that in clinical scenarios has captured over 130 resistance gene cassettes [6], although historically, most clinical class 1 integrons harbor a *dfr* gene cassette encoding resistance to trimethoprim and/or an *aad* gene cassette encoding resistance to streptomycin/spectinomycin. The class 1 integron is a highly successful, globally disseminated element. Its success can be attributed to the following points: (i) its ability to capture, arrange and express diverse resistance gene cassettes; (ii) its low fitness cost in *Enterobacteriaceae* [7]; and (iii) co-selection based on the carriage of *qacE∆1*. In most instances, class 1 integrons are central components in CRRs, although their structures continually evolve through insertion element activity [8,9] and homologous recombination [6]. Despite the bans imposed on growth promotion in many countries and improved antimicrobial stewardship practices, ARGs, biocide resistance genes and metal resistance genes are co-selected [10] in the gastrointestinal microbiomes of food animals [11,12,13], wastewater [14] and in environmental microbial populations exposed to antimicrobial residues [12,15]. Plasmids carrying complex resistance regions with combinations of resistance genes and, increasingly, virulence associated genes are well described in humans and food animals [16,17,18,19,20]. 

Sound antimicrobial stewardship practices have been evidenced in Australian agriculture by numerous reports describing *Enterobacteriaceae*, such as *E. coli,* with limited carriage of extended spectrum β-lactamase and fluoroquinolone resistance genes, and an absence of genes encoding carbapenemases [20,21,22,23]. This is consistent with the bans placed on the use of carbapenems and fluoroquinolones in animal production and judicious use of extended spectrum β-lactams. Despite this, MDR *E. coli* are a frequent occurrence in Australian intensive production systems, with the resistance genes predominantly harbored by plasmids [22,24,25]. These plasmids have persisted for long periods and have likely adapted to the diverse *E. coli* that host them [26].

Nonetheless, investigations of urban gulls and pigeons repeatedly produced *Escherichia coli* and *Salmonella enterica* that carry genes encoding resistance to fluoroquinolones, carbapenems and extended spectrum β-lactams [27,28,29,30,31]. In Australia, MDR *E. coli* have also been recovered from penguins [32] and bats [33], indicating that enterobacterial flora carried by wildlife species that intersect with human populations readily harbor ARG combinations that mirror those circulating in veterinary and human clinical environments. The *sul3* gene has been reported globally and often in association with *E. coli* sourced from intensive animal production but also frequently in humans [34]. *sul3* has also been linked with pandemic lineages of *E. coli* including ST95, encompassing various lineages renown for causing extraintestinal human (urinary tract infection, sepsis and meningitis) and poultry disease [20,35,36]. Its association with CRRs on ColV virulence plasmids, such as pCERC3 [37] in commensal *E. coli* from healthy humans [38], is concerning. *sul3*, first described in *E. coli* from swine [39], has since been found to be globally distributed in swine [40], humans [34], poultry [41,42] and wild birds [43]. The gene forms part of a *sul3*-encoding conserved segment (*sul3*-CS) that replaced the *sul1*-containing 3′-CS that is typical of class 1 integrons [37]. Several variants of the *sul3*-CS are known [34] and multiple *E. coli* sequence types have been reported to carry it, including pandemic lineages. HI2 [22,24,44] and I1 [34] plasmids are noted vehicles for the transmission of *sul3* containing class 1 integrons. Here, we describe an I1 plasmid, hosted in an *E. coli* sampled from an Australian gull, which carries a *sul3*-class 1 integron embedded within a CRR encoding an IS*26*-associated *bla*_SHV-12_ and we perform comparisons to the related plasmids hosting variants of the same CRR.

## 2. Materials and Methods

### 2.1. Isolate Sequences

The bacterial whole-genome sequence analyzed in this study has been published previously as part of a larger study of *Escherichia coli* isolates from Australian silver gulls in 2012 [45]. Briefly, isolate *E. coli* CE1867 was sourced from a cloacal sample taken from a gull chick (*Chroicocephalus novaehollandiae*) at Five Islands off the coast of New South Wales, Australia. The isolate was taken from MacConkey agar supplemented with cefotaxime (2 mg/L) and its susceptibility to a set of antimicrobials was tested as previously described [45]. Short-read whole genomes sequencing was performed on a NovaSeq Illumina platform and assembly was performed using Shovill v1.1.0 (https://github.com/tseemann/shovill, accessed on 1 November 2021). Plasmid sequence pCE1867-A is available under GenBank accession CP094826.1.

### 2.2. Reference Sequences and Metadata

Reference ST297 whole genome sequences were sourced from EnteroBase (http://enterobase.warwick.ac.uk/, accessed on 7 January 2022), along with the associated metadata, serotyping and phylotyping data [46]. A range of completed I1 plasmid sequences, selected semi-randomly to capture the range of available plasmid sequence types with as much metadata as possible, were sourced from PLSDB (https://ccb-microbe.cs.uni-saarland.de/plsdb/, accessed on 3 September 2019), along with the plasmid multi-locus sequence type data and metadata [47]. Other relevant I1 plasmid sequences were sourced from the GenBank nucleotide database (https://www.ncbi.nlm.nih.gov/nucleotide/, accessed on 2 November 2021).

### 2.3. Gene Identification, Annotation and Sequence Typing

Resistance genes were identified using ABRicate v1.0.1 (https://github.com/tseemann/abricate/, accessed on 29 October 2020) with default settings, in combination with the ResFinder database (https://bitbucket.org/genomicepidemiology/resfinder_db/src/master/, accessed on 23 December 2020) [48]. The class 1 integrase gene *intI1* (from CP059289) and insertion sequence IS*26* were identified using BLASTn (https://blast.ncbi.nlm.nih.gov/Blast.cgi, accessed on 4 November 2021), with reference sequences sourced from the GenBank nucleotide database (https://www.ncbi.nlm.nih.gov/genbank/, accessed on 4 November 2021). All other insertion sequence elements were identified using ISfinder [49]. Plasmid multi-locus sequence typing was confirmed using services from the Centre for Genomic Epidemiology (https://cge.cbs.dtu.dk/services/, accessed on 4 November 2021) [50].

### 2.4. Phylogenetics

A chromosomal single nucleotide variant (SNV) phylogeny was generated using the EnteroBase SNP project function. Plasmid sequence SNV phylogenetics were performed with parsnp v1.2 from the Harvest software suite [51] using -x (recombination filter) and -c flags, and otherwise default settings. All trees were visualized using iTOL v6.5.8 (https://itol.embl.de/, accessed on 7 January 2022) [52].

### 2.5. Data Visualization

Linear BLASTn comparisons were visualized with EasyFig v2.2.2 (https://mjsull.github.io/Easyfig/, accessed 2 September 2015) [53], with a 1000 bp size cut-off. Circular BLASTn comparisons were visualized with BRIG v0.95 (https://sourceforge.net/projects/brig/, accessed 3 January 2018) [54]. Annotations were generated and visualized using SnapGene v6.0.0.0 (https://www.snapgene.com/, accessed on 25 November 2021).

## 3. Results

Here, we describe a pST25 I1 plasmid (pCE1867-A, 115,157 bp) from *E. coli* CE1867 (ST297; phylogroup B1; serotype O45:H11), taken from a Five Islands gull, isolated on cefotaxime-supplemented media. Phenotypically, the isolate was resistant to ampicillin, streptomycin, sulphonamides, chloramphenicol, cefalotin, nalidixic acid, ceftazidime, and amoxicillin-clavulanic acid. It also encoded a *parC* E62K substitution [45]. The only other plasmid detected in the isolate was an F plasmid (F29:A-:B-), with no critical virulence factors identified. The I1 plasmid was notable for the presence of a Tn*21* transposon hosting a *sul3*-type integron with the widespread *mefB*_Δ260_ truncation (generated through IS*26* activity), plus the capture of *bla*_SHV-12,_ again through apparent IS*26* activity, and a Tn*2* transposon mobilizing *bla*_TEM-1_. 

To determine the phylogenetic placement of the host chromosome, a range of ST297 isolates (*n* = 132), selected based on the presence of the H11 *fliC* allele, were sourced from EnteroBase and placed into a SNV tree using the EnteroBase SNP project capabilities. A wide range of Shiga-toxigenic O130:H11 sequences were present within the database, primarily from plant and bovine sources in the USA and mostly positive for *stx_2_*, followed by *stx_1_* or both. Some isolates, although part of the major O130 clade, were O-non typeable or typed as D6. The isolate described here, CE1867, was a non-STEC representative of ST297 that was placed as a semi-novel clade separated from the major H11 lineage and was distinct from the other Australian isolates (Figure 1).

To determine the phylogenetic placement of plasmid pCE1867-A, a range of I1 plasmids (*n* = 70) were sourced from PLSDB for inclusion in a whole-plasmid SNV tree. ARG, *intI1* and IS*26* gene presence was determined for these plasmids and mapped alongside the phylogeny (Figure 2). These data indicated that pST69 plasmid pND11_107, isolated from a porcine *E. coli*, in the USA in 2007, was the most closely related plasmid amongst a small group of plasmids closely related to the pST2 branch of the overall phylogeny. This small group of plasmids all encoded *sul3*-type integrons with matching *psp*-*estX*-*aadA2-cmlA-aadA1* cassette profiles. Plasmids pCAZ590 (chicken, Germany) and pESBL2082-IncI (chicken, Netherlands) (MW390515.1) were each typed as pST95 and encoded *bla*_SHV-12_. In a somewhat unusual observation, a single pST3 plasmid, pMB5876 (chicken, United Kingdom) (MK070495.1) hosting this same gene profile was also identified. 

A detailed annotation of pCE1867-A alongside BRIG comparisons to closely related pND11_107 and pCAZ590, and a separate linear comparison to pND11_107 is provided in Figure 3A,B respectively. Novel regions of pCE1867-A included the Tn*2* transposon and an IS*Sso4* insertion into the I1 backbone region. 

It was noted that pCE1867-A encoded the additional sequence captured alongside *bla*_SHV-12_ that pCAZ590 was lacking within the IS*26* boundary, suggesting this transposable unit was either captured in a separate event, or that further IS*26* activity had removed part of the initially captured sequence observed in pCAZ590. A BLASTn analysis of the IS*26*-*bla*_SHV-12_-IS*26* region against the GenBank nucleotide database suggested that it is an internationally distributed element. In the case of pCE1867-A, the Tn*2* insertion has occurred into one of the bordering IS*26* elements, truncating it to 392 bp. The remaining plasmid backbone is well conserved between pCE1867-A and pND11_107, outside of the IS element activity and a short stretch of hypothetical ORFs appearing in pND11_107.

A BLASTn analysis against the GenBank nucleotide database of the Tn*21* insertion point into the I1 plasmid revealed another six plasmids that hosted Tn*21* at precisely the same location, indicating that this plasmid group has been found in Australia, the USA, France, Belgium, the Netherlands, and Germany. Details of these plasmids are shown in Table 1, all of which were sourced from agricultural samples (pigs and chickens) ranging from years 2002 to 2017, isolated from *E. coli* and *Salmonella enterica.* Plasmid sequence type data indicated that these plasmids are all from the same subclade, comprised of I1 pST25, pST26, pST69 and pST95.

Amongst the integron structures in the 11 plasmids, alterations to the integron cassette profiles did exist but were limited (Figure 4). Notably, the pST3 plasmid hosted *dfrA16* and *bla*_CARB-2_ cassettes. Several plasmids also captured *tetAR*, with p20760-1 capturing two copies. Some but not all copies of *tetAR* amongst the dataset were associated with Tn*1721.* pESBL2082-IncI (pST95) and pMB5876 (pST3) had large inversions within the complex resistance structure, with the former lacking the *intI1* gene entirely. It was notable that the *mer* operon mobilized by Tn*21* was either deleted in most examples or carried insertions of other resistance genes. Amongst the ten plasmids that were closely related (pST95, pST26, pST25, pST69), the Tn*21* insertion point was conserved with major alterations to the structure occurring at the *mer-*associated end of the transposon. The outlier to these data, pMB5876, had its insertion site close to this same position but was notably different, encoding an additional 1628 bp of the apparent plasmid backbone sequence between the Tn*21* repeat and the insertion point consistent amongst the other plasmids. Nine of the eleven plasmids all shared the *mefB*_Δ260_ deletion; however, pP136-2 harbored a full copy of *mefB* and pND11_107 carried *mefB*_Δ48_. Given the range of identical complex resistance structures presented here, and the presence of a complete *mefB* gene, the data suggest that the widely distributed *mefB*_Δ260_ truncation size originated from this plasmid lineage. Lastly, an incongruity between the plasmid phylogeny and plasmid MLST data was noted, where one pST25 plasmid was more closely related to pST69 than the two other (identical) pST25 plasmids sitting on the nearest branch. 

## 4. Discussion

Wildlife, particularly birds, are vectors for the distribution of Enterobacterial lineages that acquire ARGs on mobile genetic elements [27,55,56,57]. Despite the importance these species play in the AMR problem, our understanding of their role is still limited, constituting a major knowledge gap. In Australia, we have begun to shed light on the role played by the silver gull in the carriage and transmission of *E. coli* [28,29,30]. Recently, we identified an astonishing variety of 170 multiple drug-resistant *E. coli* lineages comprising 96 STs and representing all major phylogroups, establishing Five Islands, one of the largest breeding islands in the world, as a major site for meropenem-, cefotaxime- and ciprofloxacin-resistant *E. coli* lineages [45]. While *E. coli* lineages that display non-susceptibility to extended-spectrum β-lactams and fluoroquinolones are a hallmark of gulls sourced from different regions of Australia, lineages resistant to carbapenems so far seem restricted to samples from the Five Islands site [45]. The feeding behavior of wild bird populations lends itself to the exposure to extremely diverse enterobacterial populations found in municipal sewage plants and wastewater from hospitals, healthcare facilities and abattoirs, as well as agricultural fields carrying animal manures. Wild and urban-adapted birds are, thus, likely to acquire *E. coli* from humans and agricultural animals and then provide the opportunity for the transfer of ARGs via recombination, insertion element and integrase activity, as well as plasmid and phage transfer. It was notable to find a multiple antimicrobial resistant *E. coli* ST297 (phylogroup B1) isolate within that sample set, particularly as it was hosting an I1 plasmid with a clear agricultural association in other countries. Most of the ST297 H11 *E. coli* sequences in Enterobase are serotype O130:H11 and carry Shiga toxin genes. The gull ST297 in our study is O45:H11, lacked Shiga toxin genes, and is phylogenetically removed from other ST297 with an H11 flagella type. Based on the current data, *E. coli* ST297 (i) has a broad host range capacity, inhabiting cattle [58,59], pigs [59,60], poultry/poultry meat [61,62] and has been isolated from irrigation water [63] and food [64]; (ii) is serologically diverse; (iii) is capable of acquiring diverse plasmids and virulence genes; and (iv) is a human pathogen lineage. Collectively, these observations suggest that ST297 is a generalist lineage and a potential threat to the health of humans and animals.

Here, we have reported a comparatively benign ST297 strain that carried pCE1867-A, an I1 pST25 plasmid with the closest similarities from a phylogenomic perspective to pST69 plasmid pND11_107 [65], isolated from porcine *E. coli* from the USA in 2007. These two plasmids also cluster with pST95 plasmids pCAZ590 [66] and pESBL2082-IncI [67]. This small group of plasmids all encoded *sul3*-type integrons with matching *psp*-*estX*-*aadA2-cmlA-aadA1* cassette profiles. Plasmids pCE1867-A, pCAZ590 and pESBL2082-IncI also encoded *bla*_SHV-12_. In a somewhat unusual observation, a single more distantly related pST3 plasmid, pMB5876, hosting this same gene profile was also identified to carry *sul3* [68]. When expanding the search beyond plasmids from PLSDB, 11 I1 plasmids in total were identified in public repositories that carry *sul3*. While the majority of these I1 plasmids carry a *sul3*-class 1 integron with the globally dominant *mef**B*_Δ260_ variant, one carried a unique *mefB*_Δ48_ variant and another a full copy of the *mef**B* gene. In Australia, *sul3*-class 1 integrons have also been described in indistinguishable ColV plasmids in ST131 isolates from swine and humans [69] and pCERC3, a ColV plasmid isolated from the feces of a healthy human in Sydney [37], and mobilized by IS*26* onto HI2 plasmids in Australian swine [22]. Most of these examples are also associated with the *mef**B*_Δ260_ deletion. All the class 1 integrons described here are hosted in Tn*21,* as was originally described by Moran et al. (2016) [37]. Furthermore, all but one of these Tn*21* elements have been modified by the IS element and transposon activity, with most structures lacking the *merA* operon or with it modified in some way. The exception was pP136-2, which was additionally the only plasmid to encode a full *mefB* gene. This study strengthens the argument that the replacement of the 3′-CS of class 1 integrons by the *sul3*-CS occurred in the context of Tn*21*. A possible scenario is IntI1-mediated recombination at one end and an IS*26*-mediated event at the other to generate the *sul3*-CS, as it is commonly observed.

## 5. Conclusions

The range of isolation dates, locations, and significant variability in the presence of additional transposons and resistance genes indicates that these plasmids have been in circulation for some time, and based on the current sampling data, primarily in agricultural settings. We provide the first report of the plasmid in an avian wildlife host. This highlights the danger of AMR-encoding plasmids that circulate in economically important animal species being acquired by urban wildlife species where they may capture clinically relevant resistance genes. This observation is significant, considering Australia has always enforced strict controls on the use of clinically important antibiotics in food production systems, with the intent to prevent the introduction and persistence of genes such as *bla*_SHV-12_ in integron-based resistance structures within agricultural settings.

## Figures and Tables

**Figure 1 microorganisms-10-01387-f001:**
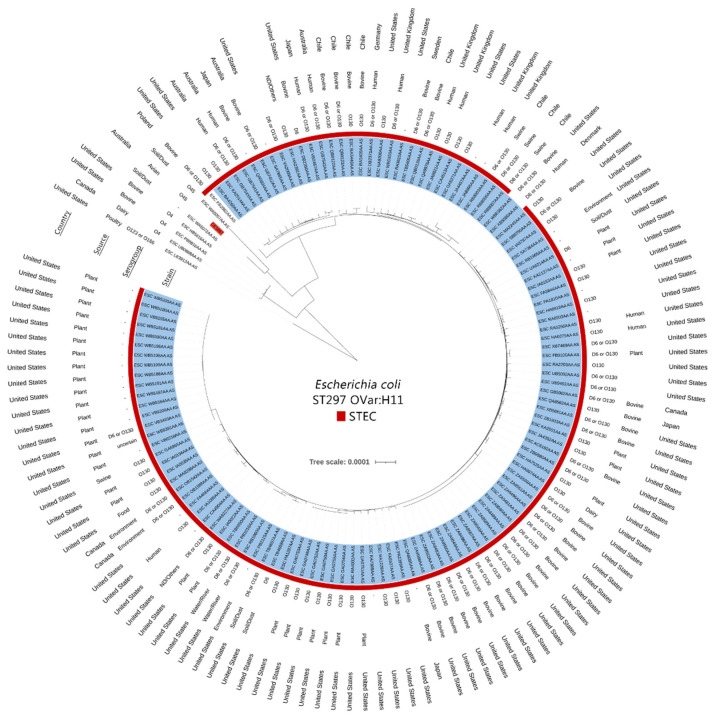
Mid-point rooted phylogenetic tree of ST297 H11 sequences. STEC O130:H11 clade is highlighted in blue, with the presence of either *stx_1_* or *stx_2_* noted as a red marker at the edge of each isolate.

**Figure 2 microorganisms-10-01387-f002:**
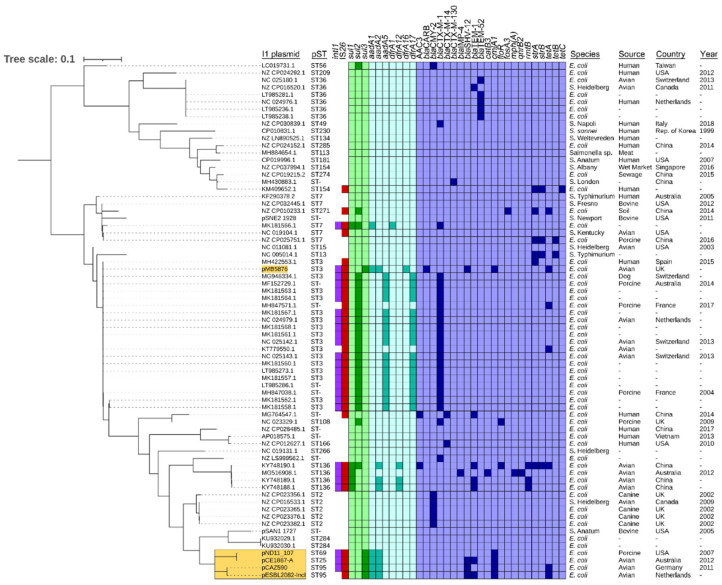
Plasmid SNV phylogeny (mid-point rooted) of I1 plasmids (*n* = 71). Data presented include plasmid multi-locus sequence typing data, presence of *intI1* and IS*26*, and ARGs, followed by available plasmid metadata (bacterial host species, isolation source, country/region, and year of isolation). Darker colors indicate a positive hit for ARGs. Plasmids relevant to this study are highlighted yellow. Tree scale is presented in SNVs per site.

**Figure 3 microorganisms-10-01387-f003:**
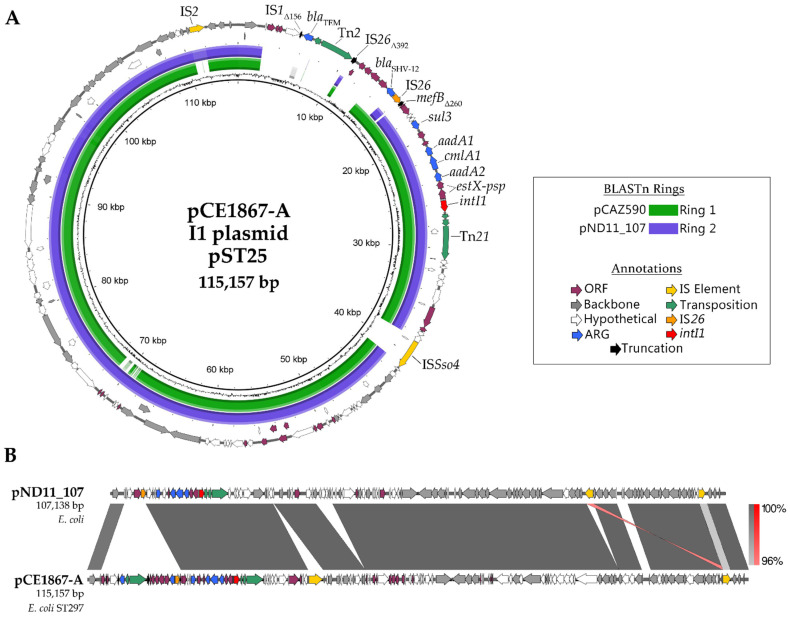
Visualized annotations and BLASTn comparisons between closely related I1 plasmids. (**A**) Circularized and annotated schematic of pCE1867-A with colored rings representing BLASTn alignments to pCAZ590 (green) and pND11_107 (blue). (**B**) Linear comparison of pND11_107 and pCE1867-A annotations, with BLASTn alignments in grey/red between.

**Figure 4 microorganisms-10-01387-f004:**
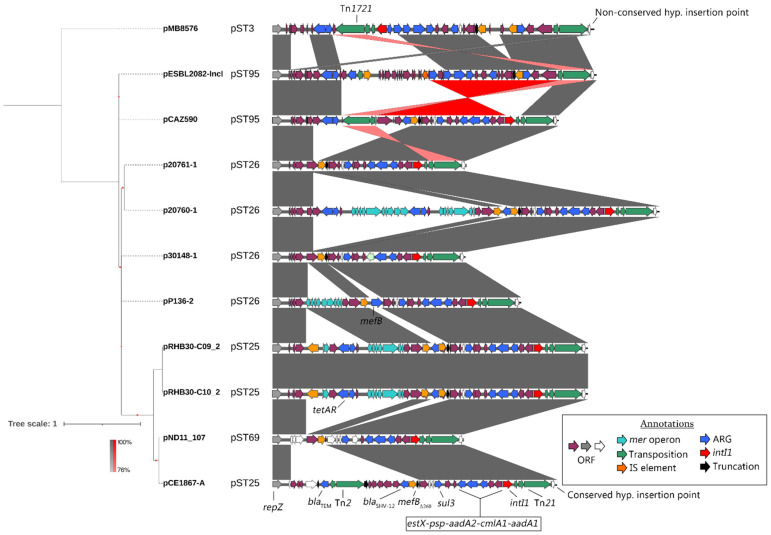
Whole-plasmid SNV phylogeny (mid-point rooted) of *sul3*-encoding I1 plasmids with linear comparisons of complex resistance regions. Tree scale is presented in SNVs per site. BLASTn similarity scale is shown in the bottom left, with red indicating an inversion of sequence.

**Table 1 microorganisms-10-01387-t001:** Details of *sul3*-encoding I1 plasmids.

Plasmid (Accession)	pST	Host	Source	Country	Isolation Date	*mefB* Size (bp)	AMR Genes	Size (bp)
pMB5876 (MK070495)	3	*E. coli*	Avian (Chicken)	Belgium	Unknown	Δ260	*aadA1*, *aadA2*, *dfrA16*, *cmlA*, *bla*_CARB-2_, *sul3*, *tetA*, *bla*_SHV-12_	117,108
pESBL2082-IncI (MW390515)	95	*E. coli*	Retail meat (Chicken)	Netherlands	Unknown	Δ260	*aadA1*, *aadA2*, *cmlA*, *sul3*, *tetA*, *bla*_TEM-1_, *bla*_SHV-12_	120,106
pCAZ590 (LT669764)	95	*E. coli* ST371	Avian (Chicken)	Germany	2011	Δ260	*aadA1*, *aadA2*, *cmlA*, *sul3*, *tetA*, *bla*_SHV-12_	117,387
p20761-1 (CP051408)	26	*S. enterica* Havana	Porcine	USA	2002	Δ260	*aadA1*, *aadA2*, *cmlA*, *sul3*	107,262
p20760-1 (CP051411)	26	*S. enterica* Heidelberg	Porcine	USA	2002	Δ260	*aadA1*, *aadA2*, *cmlA*, *sul3*, *tetA*(x2), *aph*(3′)-Ia, *mer* operon	125,574
p30148-1 (CP051354)	26	*S. enterica* Worthington	Porcine	USA	2003	Δ260	*aadA2*, *cmlA*, *sul3*	104,600
pP136-2 (CP080225)	26	*E. coli*	Porcine	France	2013	1230 (full)	*aadA1*, *aadA2*, *cmlA*, *sul3*, *mer*	103,673
pRHB30-C09_2 (CP057303)	25	*E. coli*	Porcine	UK	2017	Δ260	*aadA1*, *aadA2*, *cmlA*, *sul3*, *tetA*, *mer*	120,503
pRHB30-C10_2 (CP057294)	25	*E. coli*	Porcine	UK	2017	Δ260	*aadA1*, *aadA2*, *cmlA*, *sul3*, *tetA*, *mer*	120,503
pND11_107 (HQ114281)	69	*E. coli*	Porcine	USA	2007	Δ48	*aadA1*, *aadA2*, *cmlA*, *sul3*	107,138
pCE1869-A (CP094826)	25	*E. coli* ST297	Avian (Wild Gull)	Australia	2012	Δ260	*aadA1*, *aadA2*, *cmlA*, *sul3*, *bla*_TEM-1_, *bla*_SHV-12_	115,157

## Data Availability

Data utilized for this manuscript are available under GenBank accession CP094826 or from publicly available repositories.

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
