# Peer review of "Genomic Analysis of an I1 Plasmid Hosting a sul3-Class 1 Integron and blaSHV-12 within an Unusual Escherichia coli ST297 from Urban Wildlife"

_microorganisms, 2022, doi:10.3390/microorganisms10071387_

Round 1
Reviewer 1 Report
The authors have implemented all the recommendations of this reviewer in their manuscript and it is now ready for publication.
Reviewer 2 Report
Generally, the authors make satisfactory responses to my comments. At present, the manuscript is acceptable for publication in Microorganisms.
This manuscript is a resubmission of an earlier submission. The following is a list of the peer review reports and author responses from that submission.
Round 1
Reviewer 1 Report
The work of Wyrsch et al. reports a study in which the I1 plasmid with sul3-class 1 integron was firstly founded in an avian wildlife host. By comparing with similar plasmids, the authors demonstrated a closely related family of ARG-carrying plasmids all hosting variants of the sul3-associated integron with conserved Tn21 insertion points, and inferred that IntI1-mediated recombination at one end and an IS26-mediated event at the other to generate the sul3-CS. In generally, the results of this study were mainly from the comparisons between different plasmid sequences, which is difficult for supporting authors conclusion and can not be accepted for publication at present.
1.ESBL is the abbreviation of “Extended Spectrum Beta-Lactamase”? Please clearly write the full name.
2.“used” should be deleted in line 54, page2.
3.“Escherichia coli” should be italic type.
4.“Country” should be “Country/Region” in Fig2.
5.“Figure 3” should be “Figure 4”.
6.The authors should clearly describe the use of sulphonamide antibiotics how to influence the human and then describe its resistant genes in the second paragraph in introduction.
7.The Materials and methods, the authors should provide a more detailed description for the bioinformatics processes.
8.Authors used the whole-genome sequence from Illumina platform to assemble the plasmid, which is difficult to distinct the genes in the plasmid and bacteria genome.
9.In fig 2 and line 172-173, authors determine the phylogenetic placement of plasmid pCE1867-A by using 70 I1 plasmid from PLSDB. These plasmids were from countries or regions of Europe, Asia, Oceania and North America, not others. In table 1, sul3-encoding I1 plasmids were from more less countries (Oceania, North America and Europe). It is why that similar plasmids can not be found from other countries or regions?
10.Why 70 I1 plasmids and sul3-encoding I1 plasmids were only from Escherichia coli and Salmonella ? If the authors selected more I1 plasmids from other bacteria, it would provide more valuable information for the question of the 3'-CS of class 1 integrons how to be replaced by the sul3-CS. It is also necessary to perform the experiments to confirm the authors deduction.
Reviewer 2 Report
The manuscript of Wyrsch et al. describes the results of a bioinformatic analysis of an Escherichia coli strain and its large plasmid after an AMR screening project and whole genome sequencing. The methodology is detailed enough, the presentation of the results is of good standards and is embedded in a wide scope. Thus it confirms the importance of wildlife animals, specifically the gulls, in the dissemination and evolution of AMR plasmids. Though it is very specific and some important improvements in the text should also be made (see below).
Specific comments
The introduction is too long and contains very general and very specific statements (e.g. lanes 33-39). It can be shortened.
Lanes 48-49: something wrong with the composition.
Lanes 61-62: wrong order of sentence parts.
Lanes 84-86: wrong composition and 'multidrug-resistant'.
Lane 114: E. coli (you had the full name in lane 48 but italic).
Lane 115: E. coli (italic).
Lane 116: the gull species' name in italic.
Lane 153: omit p.
Figure 4: please, expound SNV in somewhere in the text (e.g. Materials and Methods) too.
Lanes 194-197: hardly intelligible sentences. Please, rewrite.
Reviewer 3 Report
The work entitled “Genomic Analysis of An I1 Plasmid Hosting A sul3-class 1 integron and blaSHV-12 within an unusual Escherichia coli ST297 from urban wildlife” describes the whole genome sequence of Escherichia coli isolate CE1867 from a silver gull chick sampled in 2012 that hosted an I1 pST25 plasmid with blaSHV-12, a β-lactamase gene that encodes the ability to hydrolyze oxyimino β-lactams, and other antibiotic resistance genes. The work is extremely well done and put together. The subject is well introduced, even though introduction is too extensive and should be reduced. The main goal and novelty should also be clearly identified. This is the only small point to raise in such a brilliant and interesting work. Indeed, the methodology and presentation of results is very well done and comprehensive. The discussion is also very pertinent and supported by literature. There should be, however, a separation between discussion and conclusions.